# Facilitating high quality acute care in resource-constrained environments: Perspectives of patients recovering from sepsis, their caregivers and healthcare workers in Uganda and Malawi

Felix Limbani[1]*, Olive Kabajaasi[2]*, Margaret Basemera[2], Kate Gooding[3], Nathan Kenya-Mugisha[2], Mercy Mkandawire[1], Davis Rusoke[2], Shevin T. Jacob[2,4‡], Anne Ruhweza Katahoire[5‡], Jamie Rylance[1,4‡], on behalf of the African Research Collaboration on Sepsis, Patient Experience Study Group[¶]

1 Malawi-Liverpool Wellcome Trust Clinical Research Programme, Blantyre, Malawi, 2 Walimu, Kampala, Uganda, 3 Oxford Policy Management, Oxford, United Kingdom, 4 Department of Clinical Sciences, Liverpool School of Tropical Medicine, Liverpool, United Kingdom, 5 Child Health and Development Center, Makerere University, Kampala, Uganda

‡ STJ, ARK and JR have shared senior authorship. These authors contributed equally to this work.
¶ Membership of the African Research Collaboration on Sepsis, Patient Experience Study Group is provided in the Acknowledgments.
* limbanif@yahoo.co.uk (FL); olive@walimu.org (OK)

**Data Availability Statement:** The data sets generated during and/or analysed during the

## Abstract

Sepsis is a major global health problem, especially in sub-Saharan Africa. Improving patient care requires that healthcare providers understand patients' priorities and provide quality care within the confines of the context they work. We report the perspectives of patients, caregivers and healthcare workers regarding care quality for patients admitted for sepsis to public hospitals in Uganda and Malawi. This qualitative descriptive study in two hospitals included face-to face semi-structured interviews with purposively selected patients recovering from sepsis, their caregivers and healthcare workers. In both Malawi and Uganda, sepsis care often occurred in resource-constrained environments which undermined healthcare workers' capacity to deliver safe, consistent and accessible care. Constraints included limited space, strained; water, sanitation and hygiene (WASH) amenities and practices, inadequate human and material resources and inadequate provision for basic needs including nutrition. Heavy workloads for healthcare workers strained relationships, led to poor communication and reduced engagement with patients and caregivers. These consequences were exacerbated by understaffing which affected handover and continuity of care. All groups (healthcare workers, patients and caregivers) reported delays in care due to long queues and lack of compliance with procedures for triage, treatment, stabilization and monitoring due to a lack of expertise, supervision and context-specific sepsis management guidelines. Quality sepsis care relies on effective severity-based triaging, rapid treatment of emergencies and individualised testing to confirm diagnosis and monitoring. Hospitals in resource-constrained systems contend with limitations in key resources, including for

studies in Malawi and Uganda have been uploaded as supplementary information.

**Funding:** JR and STJ received funding for this study from the UK National Institute for Health and Care Research (NIHR) using Official Development Assistance (ODA) funding [the African Research Collaboration on Sepsis, 17/63/42]. The views expressed are those of the authors and not necessarily those of the National Health Service, NIHR or the Department of Health and Social Care. JR was supported by a Wellcome Career Development Fellowship (211098Z). The funders had no role in study design, data collection and analysis, decision to publish or preparation of the manuscript. Through this NIHR funding, all co-authors received partial salary support during the period they worked on the study.

**Competing interests:** The authors have declared that no competing interests exist.

space, staff, expertise, equipment and medicines, in turn contributing to gaps in areas such as WASH and effective care delivery, as well as communication and other relational aspects of care. These limitations are the predominant challenges to achieving high quality care.

## Background

For patients with critical illness, admission to hospital is a defining moment. Multiple barriers, including complex patient pathways through pre-hospital systems, can introduce delays which are life-threatening, particularly in low-income countries (LICs) [1]. Reaching a facility, however, is only one part of the journey. For sepsis, a life-threatening condition caused by severe infection [2] responsible for 11 million deaths per year [3], rapid identification and definitive medical intervention are key to survival. Detailed recommendations for hospital-based care of patients with sepsis, including guidance on early intravenous fluid resuscitation and empiric administration of antimicrobials, are well-documented in international guidelines [4]. In contrast, other factors such as local healthcare context and environment, while rarely mentioned in such guidelines, may play an important role in the quality of care delivered and, ultimately, outcomes for patients with critical illness.

Patient safety, a component for Universal Health Coverage [5], recognizes that safe infrastructure, a skilled and committed health workforce and well-informed patients are key factors for reducing morbidity and mortality and can impact on the ability to deliver patient-centred care or enact evidence-based guidelines for sepsis. A lack of human resources, training and equipment to deal with critical illness in LICs, including in sub-Saharan Africa (sSA), has been highlighted through several quantitative analyses [6–8]. In contrast, qualitative studies that explore a patient-centred perspective on sepsis are few and mainly representative of high-income country (HIC) settings [9, 10]. Equally overlooked are interactions between healthcare staff, patients and their caregivers (i.e., family members or others who step into roles as nursing-care providers for their hospitalised relatives) in LIC settings [11]. Each of these groups have the common goal of ensuring that high-quality healthcare is delivered to maximise patient recovery. This study aimed to explore the groups' experiences, expectations and priorities regarding quality of hospital care for patients with sepsis in two LICs within sSA, Malawi and Uganda.

## Methods

### Study design

We conducted a qualitative descriptive study comprising in-person semi-structured interviews with three defined participant groups—patients recovering from sepsis, their caregivers and healthcare workers (HCWs)—in Malawi and Uganda. This qualitative study was one among several research outputs by the African Research Collaboration on Sepsis (ARCS), whose designated Centres of Sepsis Research Excellence included Malawi and Uganda. Among other objectives, ARCS aims to build capacity in sepsis research within a network of ten countries across sub-Saharan Africa. The ARCS Cohort Study aimed to determine clinical outcomes for adult patients suspected with sepsis in Malawi, Uganda, and Gabon. The qualitative study was conducted at two referral hospitals, one in Malawi (1000 beds, referral population 7 million) [12] and one in Uganda (280 beds, referral population 2 million) [13]; both are government

facilities and offer free emergency medical services and specialised care with inpatients admitted in open, single sex adult wards.

## Data collection

We purposively sampled each participant group to ensure both depth and breadth of perspectives [14]. Inclusion criteria were adult patients being discharged from hospital after recovery from sepsis; caregivers who looked after adult patients with sepsis during hospitalization and HCWs in direct and indirect care of patients with sepsis. Research staff approached unmatched patients and caregivers individually and introduced the study verbally before seeking written informed consent. Interviews were conducted in private areas (within hospital or at home) using a semi-structured interview guide, used by the interviewer to guide the interview process. We purposively selected HCWs to represent different clinical cadres (physicians, nurses, clerks and auxiliary ward staff), and we aimed for gender balance. The interviews were conducted between October 2019 and July 2020 with 61 participants across both sites (Table 1). As is standard in qualitative research, we conducted the interviews until saturation (no new information emerging from the participants). Patients and caregivers were interviewed first in the hospital and then at home one to four weeks after discharge. Audio-recorded interviews lasting approximately 1 hour were conducted by experienced social scientists (FL in Malawi; OK in Uganda) with the aid of research assistants. In Malawi, none of the participants declined to participate nor withdrew from the study. In Uganda, two patients declined participation. Although they had been discharged from the hospital, they expressed being too weak to participate in an interview.

In Malawi, data collection was concluded prior to the start of the COVID-19 pandemic. There was a slight delay in data collection in Uganda which coincided with the start of COVID-19 pandemic. For the data collection process, some participants were uncomfortable to be interviewed for fear of COVID-19 transmission; the study team addressed these concerns through assurance and adherence to COVID-19 prevention measures. As far as experience of care was concerned, participants expressed delayed access to healthcare due to interruptions in the public transport systems which were introduced in partial lockdown.

Data collection was informed by a review of the literature on quality of care, and drew on existing models of quality, including the Donabedian model and WHO framework of quality of care. Donabedian [15, 16] considers quality in terms of the *structures* (physical and organizational attributes), *processes* (interpersonal and care activities) and *outcomes* (consequences for health and well-being). WHO framework [17] considers quality in terms of *provision of care* (assessment, diagnosis and treatment, record management and referral), *experience of care* (communication and engagement, respect and emotional support) and *cross cutting areas*

**Table 1. Characteristics of study participants.**

| Participants | Malawi | | Uganda | | Total |
|---|---|---|---|---|---|
| | Male | Female | Male | Female | |
| Patients | 5 | 5 | 3 | 7 | 20 |
| Caregivers | 5 | 5 | 2 | 8 | 20 |
| HCWs | | | | | |
| Doctors | 1 | 1 | 4 | 1 | 7 |
| Nurses | 2 | 3 | 1 | 4 | 10 |
| Auxiliary ward staff | 1 | 1 | 0 | 0 | 2 |
| Receptionist/ data clerk | 1 | 1 | 0 | 0 | 2 |
| Total | **15** | **16** | **10** | **20** | **61** |

(human resource and facility environment). These frameworks overlap, for example both consider the essential physical and human resources, practices of providing care, and interpersonal communication. We drew on elements of both frameworks to identify aspects to probe within interviews, but also asked openly about experiences of care and priorities to identify factors not indicated within these frameworks

## Data management and analysis

Interviews with patients and caregivers were conducted in the local language (Chichewa in Malawi; Runyoro/Rutooro in Uganda); interviews with HCWs were conducted in English. The interviews were transcribed verbatim in the language in which the interviews were conducted and then translated into English where required, for data coding in NVIVO 12 (QSR international). We used professional translation and transcription services in MLW and Walimu, which have more than 10 years of experience of qualitative data in the medical context. FL and OK individually read through a selection of transcripts to generate an initial coding framework. Codes were reviewed by co-investigators in both sites (KG, ARK, STJ and JR). Codes were generated deductively from the study objectives and inductively as they emerged from the data. These codes were collated into themes and sub-themes and classified according to the two theoretical frameworks: the Donabedian and WHO frameworks. We examined the data for convergent and divergent patterns between groups through triangulation of perspectives. We clarified and checked issues raised in the initial interviews with participants during follow-up.

## Reflexivity statement

The lead researchers, FL and OK are local social scientists in Malawi and Uganda respectively, with previous experience doing qualitative research in health facilities. They were part of the team that designed the study considering the local context, purposively sampled participants (with the help of clinical research staff) and collected data. FL is a postdoctoral fellow and OK has MA in Sociology. Their background in social science, health systems research and better understanding of health dynamics in Africa's low-income context, helped them to better engage the patients, caregivers and HCWs. Despite this background, they undertook a naturalistic stance in the data inquiry. They studied how people experience sepsis care in the context of hospital stay in its natural environment. They acknowledge that their inherent values as researchers might have influenced the conclusion of the study, but they incorporated processes of triangulation and member checking as a way of ensuring trustworthiness. Their age and gender did not in any way affect interviewees during data collection.

## Ethics

This study was approved by Malawi's College of Medicine Research Ethics Committee (COM-REC, P.03/19/2625) and in Uganda by Makerere University School of Public Health Research and Ethics Committee (HDREC #733) and Uganda National Council for Science and Technology (#SS5162). Liverpool School of Tropical Medicine REC also approved the research as study sponsor [LSTM REC 19–020 (Malawi), 20–015 (Uganda)]. Participants were financially compensated for their time based on nationally recommended rates.

## Results

We present our findings in three broad themes developed through drawing on the Donabedian and WHO frameworks: 1) context of sepsis care; 2) experience of care taking practices

**Patient, caregiver and HCW priorities related to contexts of care**

- Nurses and doctors are adequate
- Senior medical and qualified doctors see patients in addition to interns
- Expand bed capacity and provide beddings
- Clean environment
- Improve nutrition in the hospital

**Patient, caregiver and HCW priorities related to experience of care**

- Good communication and engagement
- Patients know about their diagnosis, treatment and prognosis
- HCWs show respect and positive attitude
- Privacy during consultation and treatment
- Emotional support

**Current contexts of sepsis care (*Structures*)**

- Environmental limitations (limited space, strained hygiene and sanitation)
- Patient welfare/inadequate provision for basic needs (food)
- Providing quality medical services (inadequate staff, supplies and equipment)

**Current care taking practices (*Processes*)**

- Strained interpersonal relationships between HCWs, patients and caregivers
- Limited communication between HCWs, patients and caregivers
- Caregivers' roles
- Disparities in care provision

**Effects on sepsis care pathways (*Outcomes*)**

- Delays in triaging, treating and stabilizing patients
- Non-compliance with patient triaging, treatment and monitoring procedures as a result of:
    - *Lack of expertise, supervision and guidelines*
    - *Heavy workload*

**Patient, caregiver and HCW priorities related to provision of care**

- Appropriate examination, diagnosis and treatment
- Timely services
- Evidence-based treatment
- Free assessment and medication
- Departmental follow-up by doctors

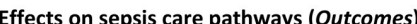

**Fig 1. Themes and sub-themes emerging from the semi-structured interviews (adapted from Donabedian model and WHO quality framework).**

and 3) effects of context and caretaking practices on sepsis care pathways (Fig 1). For each aspect, we consider the current situation, and health worker and patient priorities for improved sepsis care.

## Theme 1: Contexts of sepsis care

The limited resources for patients with sepsis in Malawi and Uganda undermine capacity to deliver safe, consistent and accessible care. Overcrowded facilities strained the existing water, sanitation and hygiene (WASH) and waste disposal facilities.

**Environmental limitations.** *Limited accommodation/space.* Inadequate space was common to both hospital settings. Resuscitation and high dependency units were congested, and ward bed numbers were frequently insufficient. In both countries, caregivers also slept on the wards, usually under or in between the beds. Limited space compromised privacy and ability to perform clinical examination and procedures and administer treatments; for example,

intravenous fluids were administered to patients lying on the floor. HCWs discharged patients early to create space for the critically ill, with patients and caregivers expressing concern that discharge was premature and occurred before sufficient recovery.

*"We ask the clinicians to at least find the patients that are better and recovering well so that they are discharged and given drugs to continue treatment at home just to create space for the new ones." **Female Nurse–Malawi***

*"They discharge us early so that new patients can get a chance to have a bed. If a patient stays here for 10 days, there will be a lot of patients sleeping along the corridors and the HCWs won't manage." **Female Patient–Uganda***

*Strained WASH amenities*. All groups were concerned about the hospital WASH facilities and practices. Participants reported incorrect disposal and sorting of wastes and sharps despite the availability of appropriate containers. Infrequent emptying of waste bins led to overflow. HCWs blamed patients and caregivers for improper use of toilets and bathrooms, including soiling and inappropriate disposal of pads, papers and food stuffs in the toilets. Cleaners reported a lack of cleaning materials (heavy duty gloves and chemicals) which limited their effectiveness. Sanitation in the hospitals was further compromised by limited and non-functional toilets and wash basins.

*"Right now, only one bathroom is being used; the other two have blockage for three weeks now. Sometimes you just go to the toilet room and use it as a bathroom." **Female Patient– Malawi***

*"My husband stopped me from going to the toilets. It seems he went there and found that they were dirty. So he would give me a bucket so that I ease myself there, and he kept taking it to the toilet to pour it." **Female Patient–Uganda***

**Patient welfare/inadequate provision for basic needs.** *Meals*. The hospital in Malawi provided complimentary patient meals. Morning porridge was considered nutritious by all groups; the other meals were viewed as nutritionally substandard, comprising *nsima* (maize staple food) and cabbage. Patients in Uganda depended on food delivered from their homes.

*"The food that is provided is not that good. The way the cabbage is prepared, it is like it was just boiled with water and salt. This is hard for people coming from far, their relatives cannot bring them food." **Female Caregiver–Malawi***

*"They should at least give us porridge. If you have come from very far, people at home may not have the money to buy you food or even come to see you. They ask you to buy drugs and you remain with no money to buy food." **Female Caregiver–Uganda***

**Quality of services.** *Insufficient staff*. HCWs reported that they were overstretched and at risk of burnout, which resulted in critically ill patients being neither identified nor monitored frequently enough.

*"We were told 1 nurse to 5–6 patients. But because we are very few its 1 nurse to 15–20 patients per day. You cannot carry out the orders (duties) or maybe you might carry out all the orders [duties] but at the end you can't have enough time to rest and refresh to go back and start helping the patients again." **Male Nurse–Malawi***

Staff shortages (doctors and nurses) compromised the time spent by clinicians with patients and the facility's capacity to attend to multiple emergencies and resulted in delays in care. Increased workloads were exacerbated by absenteeism amongst HCWs and also resulted from organizational issues (for example, hospitals dealing with patients who could be cared for in primary care facilities). HCWs wished to provide quality care to all patients, and patients expressed appreciation for those HCWs who took the time to attend to their needs.

*"Patients who are getting worse are seen frequently but identifying them in the first place is the problem. But if they are identified, they are usually followed up by both clinicians and nurses."* **Female Doctor–Malawi**

*"One clinician has to review 40 patients in the ward and each review will take ten or more minutes. . . by the time he reaches the last patient, the time for administering the medicine is gone."* **Female Nurse–Uganda**

*Shortage of medical equipment, testing and medication.* All groups noted shortages in consumables (e.g., reagents, syringes, etc.) and non-functioning equipment (e.g., laboratory analysers, x-ray, ultrasound machines) which limited the quality of care that could be delivered. Scarcity of patient monitoring equipment (e.g., thermometers, sphygmomanometers, pulse oximeters) due to breakage and theft necessitated frequent borrowing and negotiation between wards; HCWs also highlighted the limited availability of oxygen.

*"We don't have enough [oxygen] concentrators, there's only one working at the moment. So you can imagine if ten out of 60 patients are having respiratory problems. . .death will probably occur."* **Male Nurse–Malawi**

*". . .there is a patient who is very sick, needs blood gases immediately and I have to walk in the night to [another unit] and convince someone that they should share me either the syringe or the bottle of the blood gas, then I have to call their clinician on call there and convince them that their nurses should let me use their machine."* **Female Doctor–Malawi**

Staff referred carers to costly external private centres to purchase medical goods (e.g., catheter bags, intravenous cannulas). In Uganda, stock outs of pharmaceuticals were a common occurrence and resulted in delayed care.

*"These days we have not been doing investigations like LFTs [liver function tests] and CBC [complete blood count] because our machines are down. . .patients have no money to go to a private laboratory to be investigated. There are others who die before they are investigated."* **Female Nurse–Uganda**

*"I have also noted that antibiotics for organisms that are easily treated like Streptococcus pneumoniae, typhoid, are readily available. But there are instances that you get blood culture results. . .it's a multi drug resistance organism. . .we don't have those drugs so it becomes a challenge to the attending physician, it's a poor patient who cannot buy."* **Male Doctor–Malawi**

*"There is a woman who died on the ward. They told them to go and buy drugs and they failed to get the money. They waited for their relative at home to send the money and they failed. So, she died yesterday."* **Female Patient–Uganda**

### Theme 2: Experiences of care taking practices

**Interpersonal relationships between HCWs, patients and caregivers.** All groups recounted cases of unprofessional behaviour which strained interpersonal relationships and delayed treatment. Some patients and caregivers felt that HCWs were lazy, rude and non-responsive. The relationship between HCWs working in different departments could be strained, which affected handovers and created gaps in continuity of care. HCWs explained that their perceived slow approach to work was a result of the heavy workload and the unavailability of equipment and drugs. Some felt demotivated by lack of training opportunities and remuneration compounded by inequity when others received monetary top-ups.

*"Last week there was a patient who needed hourly nebulizer. Some of the nurses were doing [providing the hourly nebulizer] but some were not. . .They would say they forgot and yet the handover notes were clearly written."* **Female nurse–Malawi**

*"You can clearly see the issue of attitude. We lack commitment. Someone imagines her colleagues sleeping and she or he is here suffering and nobody is supervising them. In the end, she sleeps, she refuses to open when called [open the door when patients need them]. Others just absent themselves."* **Male Doctor–Uganda**

**Limited communication from HCWs.** Patients reported variable experience of communication, meaning that some did not understand their diagnosis and management:

*"They never explained anything to me. I just overheard them when they were doing the scanning that my kidneys are swollen. I asked what the problem was and they told me that my doctor would be the one to explain it to me. The doctor did not explain anything; he just wrote the medication I needed."* **Female patient–Malawi**

*"The nurses did a good job but this doctor was extraordinary. He would come and greet you all the time, he asks how you are, how are you feeling? What about this time? He would spend a lot of time on me."* **Male Patient–Uganda**

Some patients and caregivers wished doctors could further follow-up all their patients after referral to other departments to maintain continuity.

**Role of caregivers.** Caregivers assumed multiple roles that included explaining their patient's condition to doctors; bathing and feeding patients and sourcing food, drugs and other resources. Caregivers were frequently family members. Although minor conflicts emerged with HCWs who felt that caregiver presence occasionally compromised patient care, HCWs generally observed that patients without caregivers were disadvantaged in terms of nursing care and outcomes.

*"Another day, we were told to go out of the ward for them to mop. My patient was in pain, then she told them to call me so that I should go get drugs for her. The one who was mopping refused to open the door for me while my patient was in pain."* **Female Caregiver–Malawi**

*"There was a woman who was abandoned here. . .In such a situation when we are also few especially during evening hours, you find that you are torn apart and the workload is too much."* **Female nurse–Uganda**

**Perceived disparities in care provision.** Disparities in care delivery were sometimes perceived as partiality (e.g., being related to a nurse or a clinician or having a better socio-

economic background or status). HCWs noted that their increased attention on critically unwell patients could be misinterpreted as favouritism.

*"...we have ward rounds between 9am and 12noon, the patients that you may start with, you may have adequate time to talk to. You may also have students to teach. When the time is approaching 12 noon, it becomes difficult to see the patients adequately so we may rush things." **Male Doctor–Malawi***

*"You have to give some money to be able to beat the line. While we were there, they had withdrawn our request forms and taken them to the x-ray room. They covered them because they did not want to work without money. You would see someone who has just come is being called and helped." **Female Caregiver–Uganda***

## Theme 3: Effects of context and care taking practices on sepsis care pathways

**Delays in triaging, treating, stabilizing and monitoring patients.**   In both Uganda and Malawi, patients and caregivers described long queues during the hospital admission process. In Malawi, patients and caregivers questioned whether the reasons for long delays during triage were related to heavy workload or HCWs' work ethic.

*"...they take a long time to check on a patient...they must check patients as soon as they arrive to see if there's need for urgency." **Female Caregiver–Malawi***

*"We arrived at the reception at 9:00 am and I stayed in the line for 3 hours, yet I was very sick. Those giving numbers are slower than the time you spend when the doctor is seeing you in OPD (Out-patient department)." **Female Patient–Uganda***

Other patients described positive experiences of being rapidly attended, contrary to their expectations and consistent with good triage processes:

*"At first, when we reached and found a line, I said . . . "Oh God", I used to hear people talk about [hospital name], that one can die while on the queue. But that is not what I found. I went straight and when I reached the ward, doctors came and worked on me immediately." **Male Patient–Uganda***

While HCWs identified systems issues such as multiple parallel emergencies and protracted hand-overs between departments, patients and caregivers pointed to staff shortage and long lunch breaks. Additionally, laboratory delay and suboptimal prioritisation of blood testing and reporting frustrated all groups.

*"A patient sent to the lab, they stay there the whole day and they come back and tell you 'I have not been worked on'. The lines are always long; they don't prioritize very sick patients on ward." **Female Nurse–Uganda***

*"Samples are coming from different departments, but the staff could be helping us by preparing the samples marked 'emergency', faster." **Male Nurse–Malawi***

Delays were also compounded by the inherent limitations of some tests for sepsis.

*"After 5 o'clock, people are not happy to do a blood culture, that's what I think. Unfortunately, blood cultures even if you take them correctly, they will not always show you what bacteria it is. . .if you are lucky, it will come back in 72 hours, probably by this time you already realized that the antibiotic you gave them is not working."* **Female Doctor–Malawi**

All groups reported experiencing misplaced or lost blood samples as a contribution to delayed treatment. In some cases, HCWs said they failed to remember where they placed the samples, and caregivers in both Malawi and Uganda explained that HCWs did not mark the collection bottles properly or that laboratory reports were misplaced or lost; subsequent need for repeat blood testing was received with resistance by both patients and caregivers.

*"They have taken blood from him 5 times. It looks like the patient is donating blood, because they come without telling us the reason why they are taking blood samples, what they want to test, and they don't even give us report of the outcome. . ."* **Male Caregiver–Malawi***

(*For context, rumours of malevolent "blood suckers" who attack individuals at night are commonplace in Malawi)

*"I got angry and asked him, 'why are you saying that I should go and test more blood?'. He said that they lost the previous lab report. I thought that every time we bring a copy, you put it in the file. I told them, 'If you keep telling me to go out and repeat tests, it shows that I'm equally responsible for the custody of my patient's files', which is wrong."* **Female Caregiver–Uganda**

**Non-compliance with patient triaging, treatment and monitoring procedures.** *Lack of expertise, supervision, and guidelines.* HCWs reported that procedures for quality management of patients admitted with sepsis in accordance with existing guidance were not carried out due to lack of knowledge regarding sepsis management or lack of documentation.

*"Sometimes people don't do all the vital signs; they may do the blood pressure, but without checking the temperature. So that's the first challenge, one identifying that this patient has got sepsis using the set standards. Secondly, knowing what to do at the right time."* **Male Doctor–Malawi**

Patients and caregivers expressed limited confidence in trainees, questioning interns and some HCWs for failure to perform certain clinical procedures. They preferred to be attended to by senior doctors. HCWs in Uganda suggested that guidelines for sepsis would help address this issue:

*"At the policy level, I may say that we do not have algorithms for managing patients with sepsis in this hospital. . .. . . algorithms give you step by step way of how the patients should be managed until when they are discharged."* **Male Doctor–Uganda**

*High workload affecting patient monitoring.* Patients reported missed doses of prescribed medications due to HCWs' lack of time or oversight ascribed to staff shortages. Clinicians agreed that these omissions were detrimental to care:

*"The doctor may prescribe, ok this patient give them a bolus of IV fluids and maybe continue with 3 litres over 24 hours. He will come there the following day and find that the patient only*

*received that 1 litre that was put in [the emergency department]; the reason is maybe there is no one to continue with monitoring."* **Male Doctor–Malawi**

*"Much they are seen and prescriptions are made on a daily basis, timing of administering the medicine is the biggest problem. If the patient is not receiving what you have written, then it is really nothing there."* **Male Doctor–Uganda**

In some cases, although the patients were seen and prescription were made on a daily basis, timing of administering the medicine was the biggest problem:

*"There was one doctor who was working on day duty. I told them that the drip is finished and told me that I go and tell another doctor to come and change the drip. But all the people I went to said they were very busy."* **Female Caregiver–Uganda**

HCWs acknowledged gaps in patient monitoring. They tried to monitor very sick patients at least twice a day. Caregivers also felt responsibility for timely management of their patients and prompted nurses.

*"Caregivers come to us to complain about IV fluids. . .If the drugs are missed they always tell you that, doctor prescribed this drug yesterday, received an injection, but today has not received any injection, we ask the nurses or the doctors they say the medication is not available, it is out of stock so it means that day is missed."* **Male Doctor–Malawi**

HCWs understood these problems contributed to delays in care and tried to mitigate against them. For example, to help detect deterioration sooner, very sick patients slept closer to the nurses' station for ease of monitoring, which was positively perceived by caregivers.

*"Those that have severe infections like septicaemia, severe malaria, cardiac failure and others are reviewed twice daily depending on the situation really. They even have special beds, if you look this side of female ward and even male ward, those patients sleep on beds closer to the nursing station."* **Male Doctor–Uganda**

### Priorities for improving quality of sepsis care

Interview participants were asked to describe their top three priorities for improving quality of care. Additionally, HCWs were asked to describe what they felt were priorities of patients. Priorities presented were those common across Malawi and Uganda (Table 2). We observed that the priorities were related to issues that participants mentioned as problems.

All groups prioritised improvements in timely, evidence-based, and free assessment, diagnosis and treatment; having adequate human resource and improving accommodation, WASH facilities and nutrition in the hospitals. Patients and caregivers preferred better communication and engagement with HCWs. HCWs felt that respect, privacy, and support from HCWs are some of the priorities of patients. We noted that the different priorities identified by patients/caregivers and HCWS, were all within the dimension of "Experience of Care". Despite having this as a priority across the groups, it presents the different viewpoints that patients/caregivers and HCWs have in areas of communication, respect, and emotional support. This provides an opportunity for further research to explore how to reconcile these viewpoints and promote patient-provider interaction.

**Table 2. Priorities in improving the quality of sepsis care from the perspective of patients with sepsis, their caregivers and HCWs aligned with elements of the WHO standards for improving quality of maternal and neonatal care.**

| Dimension | WHO standards of care | Priority areas for improving quality of sepsis care | Patients | Caregivers | HCWs |
|---|---|---|---|---|---|
| Provision of care | Assessment, diagnosis and treatment | Getting required examination, appropriate diagnosis, treatment and getting healed | X | X | X |
| | | Timely services. Patients are against waiting for long periods of time | X | X | X |
| | | Evidence-based treatment, in relation to their condition, should be readily available for patients | X | X | X |
| | | Receiving free assessment and medication using modern equipment without being told to do the tests outside of the hospital | X | X | X |
| | Referral | Doctors being able to follow up with all their patients when they have been referred to other departments | X | X | |
| Experience of care | Communication and engagement | Feeling welcome at the hospital and given adequate attention. | X | X | |
| | | Receiving explanation of the results of their tests and being given confidence that they are getting the right treatment (building trust) | X | X | |
| | Respect | Patients prefer HCWs that have a positive attitude and show respect; being addressed in a polite way | | | X |
| | | Privacy during the course of patient's consultation and treatment | | | X |
| | Emotional support | Receiving support and comfort from friends and relatives | X | X | |
| | | Caregivers always wish to be in the wards, closer to their patients and see how their patient is being treated | | | X |
| Cross cutting areas | Human resource | Nurses and doctors to be readily available whenever patients call for help | X | X | X |
| | | Patients would rather be seen by a senior medical and qualified doctor than by interns | X | X | X |
| | Facility environment | Improve on accommodation including expanding the bed capacity/ wards areas and providing beddings | X | X | X |
| | | A clean environment: ensuring that bathrooms and toilets are not dirty; consistent supply of water and health education on hygiene | X | X | X |
| | | Improve nutrition in the hospital, by providing a variety of well-prepared food | X | X | X |

## Discussion

This study described the context in which sepsis care is provided in two public hospitals in LIC settings within sSA. In both Malawi and Uganda, sepsis care involved dynamic relationships between patients, caregivers and HCWs and required coping with significant limitations of space, WASH amenities, staff, equipment and medicines. These resource constraints contributed to interpersonal tensions between HCWs, patients and their caregivers, with heavy workloads of HCWs reducing the quality of engagement and communication. Moreover, care pathways for patients with sepsis were destabilised in the context of compounded triage delays and compromised clinical assessments, ultimately resulting in suboptimal treatment.

Our findings highlight what issues matter to HCWs, patients and caregivers in the care of critical illnesses like sepsis. In common with other critical conditions, sepsis care is likely to be impacted most in environments where health systems have critical shortage of medical resources, including staff. Initiatives for preventing sepsis place emphasis on WASH in both communities and care facilities. We have noted multiple reasons for compromised ward hygiene, including inadequate and non-functional sanitation amenities and unhygienic practices by the three groups. Cultural practices in Malawi and Uganda reflect the interrelated roles that communities and hospitals play in caring for the sick. Strengthening interpersonal relations is essential in crossover of responsibilities and proximity among patients, caregivers and HCWs.

Our findings confirm observations from other LIC settings that include challenges of limited bed space; shortages of staff; poor WASH and infection prevention and control practices

[1, 18, 19]; insufficient diagnostic capacity; inadequate treatment, including intravenous fluids and supplemental oxygen, and shortage of monitoring equipment, such as pulse oximeters [1, 7, 20]. Knowledge and skills gaps regarding diagnostic criteria and management of sepsis are perpetuated by lack of context-specific guidelines [7, 8, 20, 21].

In other studies of hospitalised patients in LIC settings, behaviour and relationships have also been noted as important aspects of care [22, 23]. A scoping review of communication in nurse-patient interaction in sub-Saharan Africa has noted poor communication, where nurses dominate the process, neglect patient needs and concerns, and abuse and humiliate patients. Such behaviours have been influenced by excessive workload, shortages of nursing staff, and poor communication skills [24, 25]. Studies in HIV and AIDS, maternal and reproductive health, intensive and palliative care, and primary health care have documented these behaviours. Although caregivers play significant roles in the care of inpatients, they are subjected to poor accommodation and loss of income during their hospital stay [23]. Poorly organized handovers and limited communication between HCWs and caregivers limit the continuity of acute care [26]. Hospital care is also affected by high levels of HCW absenteeism which results in work overload and stress for the remaining HCWs [27].

Our study highlighted additional factors which impact on sepsis care, including long patient waiting times, limited explanations of patients' diagnosis and prognosis, disparities in provision of care and lack of nutritional meals. Those few studies which have documented hospital experiences of patients with sepsis, their caregivers and HCWs derive from HIC settings [9, 10]. The studies about sepsis in LIC settings are mostly specific to one group of participants, a section of the hospital or a component of sepsis care. Our study brings together the common experiences of patients, caregivers and HCWs in both Malawi and Uganda, countries with similar but distinct health systems. Taken together with emergency care literature, our findings suggest how these experiences may be broadly applicable across resource-constrained settings.

Additional strengths of this study include the value of follow-up interviews with patients and caregivers in their homes, which accorded them time to reflect on their hospital experience. Using the two theoretical framework approaches helped to generate comparative data across the two study sites and strengthened the methodological approach in this study. While the Donabedian framework placed the findings into the three broad categories of care, the WHO framework complemented it by further breaking it down into priority standards of care. We note that we conducted this study alongside a prospective cohort study measuring clinical outcomes of patients with sepsis. Additional care including in-hospital and out-patient follow-up visits might have influenced our respondents' perceptions of care.

Interventions to improve service quality are likely to require significant investment, but such investment is not always financial. Education interventions can improve compliance to medication prescription by HCWs [28]. Other studies in LMICs, such as ICU experience with critically ill children and patients, have demonstrated that close monitoring and excellent communication are possible and translate to high satisfaction with the level of care [19, 22]. Our study highlights how priorities for patients, caregivers and HCWs can be translated into interventions for improving the quality of hospital care for patients with sepsis in resource-constrained settings.

## Conclusion

In the context of resource-constrained health systems, our findings show how the basic care pathway is destabilized due to structural and relational challenges. Hospitals have to deal with limitations in space, WASH, staff, expertise, equipment, medicines and nutrition. Although interactions between HCWs, caregivers and patients are important in these two settings,

improving hospital-level resources could strengthen engagement and communication between these groups and facilitate high quality care for critical illnesses like sepsis.

## Supporting information

**S1 Text. The African Research Collaboration on Sepsis, Patient Experience Study Group.** (DOCX)

**S1 File. Experience of sepsis care (data for Malawi).** (DOCX)

**S2 File. Experience of sepsis care (data for Uganda).** (DOCX)

## Acknowledgments

We appreciate the contributions and participation of the rest of the members of the African Research Collaboration on Sepsis, Patient Experience Study Group. These include Barbara Njamwaha[1], Hilda Muwando[2], Jacob Phulusa[1], Lucy Keyala[1], Madalitso Chiutsi[1], Priscilla Haguma[2], Sharon Nyesiga[2], Sylvester H. Kaimba[1] and Solomon Kyakuha[3].

[1] Malawi-Liverpool Wellcome Trust Clinical Research Programme, Chichiri, Blantyre 3, Malawi

[2] Walimu, Coral Crescent, Kololo, Kampala, Uganda

[3] Hoima Regional Referral Hospital, Uganda

## Author Contributions

**Conceptualization:** Kate Gooding, Shevin T. Jacob, Anne Ruhweza Katahoire, Jamie Rylance.

**Data curation:** Felix Limbani, Olive Kabajaasi, Margaret Basemera, Nathan Kenya-Mugisha, Mercy Mkandawire, Davis Rusoke.

**Formal analysis:** Felix Limbani, Olive Kabajaasi, Kate Gooding, Shevin T. Jacob, Anne Ruhweza Katahoire, Jamie Rylance.

**Funding acquisition:** Shevin T. Jacob, Jamie Rylance.

**Investigation:** Felix Limbani, Olive Kabajaasi.

**Methodology:** Felix Limbani, Olive Kabajaasi, Shevin T. Jacob, Anne Ruhweza Katahoire, Jamie Rylance.

**Project administration:** Felix Limbani, Olive Kabajaasi.

**Resources:** Shevin T. Jacob, Jamie Rylance.

**Supervision:** Shevin T. Jacob, Anne Ruhweza Katahoire, Jamie Rylance.

**Validation:** Felix Limbani, Olive Kabajaasi.

**Writing – original draft:** Felix Limbani, Olive Kabajaasi, Kate Gooding, Anne Ruhweza Katahoire.

**Writing – review & editing:** Felix Limbani, Olive Kabajaasi, Margaret Basemera, Kate Gooding, Nathan Kenya-Mugisha, Mercy Mkandawire, Davis Rusoke, Shevin T. Jacob, Anne Ruhweza Katahoire, Jamie Rylance.

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
