## [Decision Letter · Decision Letter 0]

5 May 2022

PGPH-D-22-00087

Facilitating high quality acute care in resource-constrained environments: perspectives of patients recovering from sepsis, their caregivers and healthcare workers in Uganda and Malawi

Dear Dr. Limbani,

Thank you for submitting your manuscript to PLOS Global Public Health. After careful consideration, we feel that it has merit but does not fully meet PLOS Global Public Health’s publication criteria as it currently stands. Therefore, we invite you to submit a revised version of the manuscript that addresses the points raised during the review process. My comments are included in the reviews below, but in general, please make sure that reporting adheres closely to established guidlelines for qualitative research such as COREQ or similar. It would also be useful if you could explore further how your findings on professionalism and staff communication/behaviour (and it's association with overwork/potential burnout) relates to the exisiting literature in acute care - these are important findings that could be developed more.

Please submit your revised manuscript by . If you will need more time than this to complete your revisions, please reply to this message or contact the journal office at globalpubhealth@plos.org. Please include the following items when submitting your revised manuscript:

We look forward to receiving your revised manuscript.

Kind regards,

M. Dylan Bould

Academic Editor

Journal Requirements:

2. Please update your Competing Interests statement. If you have no competing interests to declare, please state: “The authors have declared that no competing interests exist.”

Additional Editor Comments (if provided):

I really enjoyed reading this study. It deals with an important research question and the qualitative methods are well suited to that question. It is a strength that data comes from two countries, it would be interesting to know how these countries happened to be selected for this study.

I think it’s really important that the reporting of this study adheres to current guidelines, e.g. COREQ https://www.equator-network.org/reporting-guidelines/coreq/ or SRQR https://www.equator-network.org/reporting-guidelines/srqr/ or both. Please go through the checklists and the associated guidelines in detail and ensure that you have considered carefully whether your reporting aligns with each point. In particular, consider points on relationship with patients, reflexivity, general methodological orientation, non-participation, setting of data collection, interview guide, field notes, data saturation/thematic sufficiency, participant checking.

I agree with one of the reviewers that it is particularly important to note that the data collection period included the start of the COVID pandemic - and to comment on whether your interviews and findings were influenced by this. In fact, COVID is not mentioned in your study - which is somewhat surprising.

Purposive sampling is appropriate. I would like to know which of the research staff involved in the study were responsible for selecting the participants, as part of reflexivity. It would also be interesting to have some idea about the denominator - of how many healthcare professionals and/or patients were available for recruitment over the period of time of the study.

Your inclusion criteria was adults who were discharged from the hospital after recover from sepsis. What of patients who failed to recover from sepsis - can you please comment on this?

Who translated the transcripts? Did you use any software for the organization and analysis of your data? Thanks for noting the existing theoretical frameworks that influenced your deductive analysis of the data, can you also comment how you would characterize your analysis in terms of general qualitative paradigms/methodological orientation - e.g. some form of content analysis, grounded theory, qualitative case study etc. I think it would also be useful if you commented on your epistemological stance, e.g. was your position post-positivist, social constructivist etc?

Does the “WASH” acronym apply to “space, strained water, sanitation and hygiene”? the letters don’t seem to quite fit.

The data contains rich descriptions of the challenges to effective care for patients with sepsis. These findings are important and often resonate with what I’ve seen in other institutions in East and Southern Africa. Some of the descriptions of challenging behaviour and communication by healthcare workers are the the most troubling and difficult data in the paper. Please expand on whether similar findings are found elsewhere in the literature, not necessarily for care of patients with sepsis, but in other acute care contexts - I think it’s important to situate these findings within a broader context.

Reviewers' comments:

Reviewer's Responses to Questions

**Comments to the Author**

1. Does this manuscript meet PLOS Global Public Health’s publication criteria? Is the manuscript technically sound, and do the data support the conclusions? The manuscript must describe methodologically and ethically rigorous research with conclusions that are appropriately drawn based on the data presented.

Reviewer #1: Yes

Reviewer #2: Yes

2. Has the statistical analysis been performed appropriately and rigorously?

Reviewer #1: I don't know

Reviewer #2: Yes

3. Have the authors made all data underlying the findings in their manuscript fully available (please refer to the Data Availability Statement at the start of the manuscript PDF file)?

Reviewer #1: Yes

Reviewer #2: Yes

4. Is the manuscript presented in an intelligible fashion and written in standard English?

Reviewer #1: Yes

Reviewer #2: Yes

5. Review Comments to the Author

Reviewer #1: Thank you for the opportunity to review the manuscript of this qualitative study examining sepsis care perspectives and priorities among patients, caregivers and healthcare providers in Uganda and Malawi.

This research addresses important knowledge gaps in the current state of sepsis care provision in resource-constrained settings. The authors used two relevant quality of care frameworks to guide data collection and analysis. The included interview excerpts were both candid and informative. The study also clearly highlights priority areas for future focus and investments.

I have some comments to be considered:

1. Methods: One referral hospital from each country was selected as the study site. Why were these two referral hospitals selected specifically? Are they representative of other hospitals that provide critical care in the country? The authors could elaborate on the differences between government versus other facilities to provide more context and rationale.

2. Methods: Were there any participant exclusion criteria? Were any caregivers of patients who did not recover from sepsis included? These caregivers may have unique insights to provide to the study.

3. Results: The authors can consider providing more demographic information about the patients and caregivers, e.g., age and length of hospital stay, to provide further context for the data.

4. Discussion: Given that there were some differences between the priorities identified by patients/caregivers and healthcare workers, the authors can consider addressing how to reconcile or explore this further.

5. Discussion: The authors can consider a discussion of limitations of the current study.

Minor comments

- Please provide description of the acronym “OPD” on page 15 line 297

- Please provide a higher resolution image of Figure 1

Reviewer #2: Thank you,

The authors present results of a qualitative study conducted among a population derived from a sepsis ecosystem. This included patients recovering from sepsis, their caregivers and healthcare workers who were part of a sepsis study cohort.

The authors must be commended for attempting to address a poorly understood facet of sepsis care in Low income settings.

My comments are minor in general.

1. The abstract while without headings, should nonetheless follow the IMRD format. Bits of the introduction are after the methods. It maybe a tense issue, but it does not flow clearly.

2. The background reads well.

3. In the methods section, the study settings need a little more detail to prove similarity. In one hospital (in Malawi, they seem to have some access to blood gases and cultures, which the Ugandan hospital does not have, does this imply a better equipped Malawian hospital and does this then not risk bias?)

4. In qualitative work, thematic saturation usually determines the final sample size. Did the cut off cover all three domains, or just one? This needs clarity.

5. What software was used to transcribe the recordings? What skillsets and experience did the transcribers have. What quality assurance was conducted to ensure agreement in thematic saturation?

6.Wouldn't having just 2 sites (hospitals) be a study limitation? While I agree with the findings being a LMIC practitioner? I am concerned there may be some generalisability issues. This can be addressed as a possible study limitation.

7. The absence of treatment algorithms for sepsis can also be discussed within the context of the WHO-IMAI guidelines which were rolled out to SSA ministries of health. Is this not an implementation issue?

8. Did the study occur at similar pace in the different countries? The COVID pandemic was well and truly underway, how did the lockdowns and surges affect the study.

6. PLOS authors have the option to publish the peer review history of their article (what does this mean?). If published, this will include your full peer review and any attached files.

**Do you want your identity to be public for this peer review?** For information about this choice, including consent withdrawal, please see our Privacy Policy.

Reviewer #1: **Yes: **Jenny Hoang Nguyen

Reviewer #2: No

---

## [Decision Letter · Decision Letter 1]

13 Jul 2022

Facilitating high quality acute care in resource-constrained environments: perspectives of patients recovering from sepsis, their caregivers and healthcare workers in Uganda and Malawi

PGPH-D-22-00087R1

Dear Dr Limbani,

We are pleased to inform you that your manuscript 'Facilitating high quality acute care in resource-constrained environments: perspectives of patients recovering from sepsis, their caregivers and healthcare workers in Uganda and Malawi' has been provisionally accepted for publication in PLOS Global Public Health.

Best regards,

M. Dylan Bould

Academic Editor

Reviewer Comments (if any, and for reference):

Reviewer's Responses to Questions

**Comments to the Author**

1. If the authors have adequately addressed your comments raised in a previous round of review and you feel that this manuscript is now acceptable for publication, you may indicate that here to bypass the “Comments to the Author” section, enter your conflict of interest statement in the “Confidential to Editor” section, and submit your "Accept" recommendation.

Reviewer #2: All comments have been addressed

2. Does this manuscript meet PLOS Global Public Health’s publication criteria? Is the manuscript technically sound, and do the data support the conclusions? The manuscript must describe methodologically and ethically rigorous research with conclusions that are appropriately drawn based on the data presented.

Reviewer #2: (No Response)

3. Has the statistical analysis been performed appropriately and rigorously?

Reviewer #2: (No Response)

4. Have the authors made all data underlying the findings in their manuscript fully available (please refer to the Data Availability Statement at the start of the manuscript PDF file)?

Reviewer #2: (No Response)

5. Is the manuscript presented in an intelligible fashion and written in standard English?

Reviewer #2: (No Response)

6. Review Comments to the Author

Reviewer #2: (No Response)

7. PLOS authors have the option to publish the peer review history of their article (what does this mean?). If published, this will include your full peer review and any attached files.

**Do you want your identity to be public for this peer review?** For information about this choice, including consent withdrawal, please see our Privacy Policy.

Reviewer #2: **Yes: **Arthur Kwizera
